# A Comparison of Hamming Errors of Representative Variable Selection Methods

**Zheng Tracy Ke**
Department of Statistics
Harvard University
Cambridge, MA 02138, USA
zke@fas.harvard.edu

**Longlin Wang**
Department of Statistics
Harvard University
Cambridge, MA 02138, USA
lwang2@fas.harvard.edu

## Abstract

Lasso is a celebrated method for variable selection in linear models, but it faces challenges when the variables are moderately or strongly correlated. This motivates alternative approaches such as using a non-convex penalty, adding a ridge regularization, or conducting a post-Lasso thresholding. In this paper, we compare Lasso with 5 other methods: Elastic net, SCAD, forward selection, thresholded Lasso, and forward backward selection. We measure their performances theoretically by the expected Hamming error, assuming that the regression coefficients are *iid* drawn from a two-point mixture and that the Gram matrix is block-wise diagonal. By deriving the rates of convergence of Hamming errors and the phase diagrams, we obtain useful conclusions about the pros and cons of different methods.

## 1 Introduction

Variable selection is one of the core problems in high-dimensional data analysis. Consider a linear regression, where the response $y \in \mathbb{R}^n$ and the design matrix $X = [x_1, \ldots, x_p] \in \mathbb{R}^{n \times p}$ satisfy that

$$y = X\beta + z, \qquad \|x_j\| = 1, \qquad z \sim \mathcal{N}(0, \sigma^2 I_n). \tag{1}$$

The goal is estimating the support of $\beta$ ($\mathrm{Supp}(\beta)$). Lasso (Tibshirani, 1996) is a popular method:

$$\hat{\beta}^{\mathrm{lasso}} = \mathrm{argmin}_\beta \big\{ \|y - X\beta\|^2/2 + \lambda \|\beta\|_1 \big\}. \tag{2}$$

Lasso has good rates of convergence on the $L_q$-estimation error or prediction error (Bickel et al., 2009). However, it can be unsatisfactory for variable selection, especially when the columns in the design matrix are moderately or strongly correlated. Zhao & Yu (2006) showed that an *irrepresentable condition* on $X$ is necessary for Lasso to recover $\mathrm{Supp}(\beta)$ with high probability, and such a condition is restrictive when $p$ is large (Fan & Lv, 2010). Ji & Jin (2012) studied the Hamming error of Lasso and revealed Lasso's non-optimality by lower-bounding its Hamming error rate. Many alternative strategies were proposed for variable selection, such as using non-convex penalties (Fan & Li, 2001; Zhang, 2010; Shen et al., 2012), adding a ridge regularization (Zou & Hastie, 2005), post-processing on the Lasso estimator (Zou, 2006; Zhou, 2009), and iterative algorithms (Zhang, 2011; Donoho et al., 2012). In this paper, our main interest is to theoretically compare these different strategies.

Existing theoretical studies focused on 'model selection consistency' (e.g., Fan & Li (2001); Zhao & Yu (2006); Zou (2006); Meinshausen & Bühlmann (2010); Loh & Wainwright (2017)), which uses $\mathbb{P}(\mathrm{Supp}(\hat{\beta}) = \mathrm{Supp}(\beta))$ to measure the performance of variable selection. However, for many real applications, the study of the Hamming error (i.e., total number of false positives and false negatives) is in urgent need. For example, in genome-wide association studies (GWAS) or Genetic Regulatory Network, the goal is to identify the genes or SNPs that are truly associated with a given phenotype, and we hope to find a multiple testing procedure that simultaneously controls the FDR and maximizes the power (for multiple testing). This problem can be re-cast as minimizing the Hamming error in a special regression setting (Efron, 2004; Jin, 2012; Sun & Cai, 2007). This motivates us to study the *Hamming errors* of variable selection methods, which were rarely considered in the literature.

We adopt the *rare and weak signal model* (Donoho & Jin, 2004; Arias-Castro et al., 2011; Jin & Ke, 2016), which is often used in theoretical analysis of sparse linear models. Let $p$ be the asymptotic

parameter. Given constants $\vartheta \in (0, 1)$ and $r > 0$, we assume that $\beta_j$'s are iid generated such that

$$\beta_j = \begin{cases} \tau_p, & \text{with probability } \epsilon_p, \\ 0, & \text{with probability } 1 - \epsilon_p, \end{cases} \qquad \text{where} \qquad \epsilon_p = p^{-\vartheta}, \quad \tau_p = \sqrt{2r \log(p)}. \qquad (3)$$

As $p \to \infty$, $\|\beta\|_0 \approx p^{1-\vartheta}$, and a nonzero $\beta_j$ is at the critical order $\sqrt{\log(p)}$. [1] The two parameters $(\vartheta, r)$ capture the sparsity level and signal strength, respectively. We may generalize (3) to let nonzero $\beta_j$'s take different values in $[\tau_p, \infty)$, but the current form is more convenient for presentation.

The *blockwise covariance structure* is frequently observed in real applications. In genetic data, there may exist strong correlations between nearby genetic markers, but the long-range dependence is usually negligible; as a result, the sample covariance matrix is approximately blockwise diagonal (Dehman et al., 2015). In financial data, the sample covariance matrix of stock returns (after common factors are removed) is also approximately blockwise diagonal, where each block corresponds to an industry group (Fan et al., 2015). Motivated by these examples, we consider an idealized setting, where the Gram matrix $G = X'X$ is block-wise diagonal consisting of $2 \times 2$ blocks:

$$G = \text{diag}(B, B, \ldots, B, B_0), \qquad \text{where} \quad B = \begin{bmatrix} 1 & \rho \\ \rho & 1 \end{bmatrix} \text{ and } B_0 = \begin{cases} B, & \text{if } p \text{ is even}, \\ 1, & \text{if } p \text{ is odd}. \end{cases} \qquad (4)$$

This is an idealization of the blockwise covariance structures in real applications. We may generalize (4) to allow unequal block sizes and unequal off-diagonal entries, but we keep the current form for convenience of presentation. Model (4) is also closely connected to the random designs in compressed sensing (Donoho, 2006). Write $X = [X_1, X_2, \ldots, X_n]'$. Suppose $X_1, X_2, \ldots, X_n$ are iid generated from $\mathcal{N}(0, n^{-1}\Sigma)$, where $\Sigma$ has the same form as $G$ in (4). In a high-dimensional sparse setting, we have $\|\beta\|_0 \ll n \ll p$. Then, $G = X'X \approx \Sigma$, and due to the blessing of sparsity of $\beta$, $G\beta \approx \Sigma\beta$. As a result, $X'y$ (sufficient statistic of $\beta$) satisfies that $X'y = G\beta + \mathcal{N}(0, G) \approx \Sigma\beta + \mathcal{N}(0, \Sigma)$, and the right hand side reduces to Model (4) (Genovese et al., 2012). In Section 3.5, we formally show that this random design setting is asymptotically equivalent to Model (4).

Now, under model (3) and model (4), we have three parameters $(\vartheta, r, \rho)$. They capture the sparsity level, signal strength and design correlations, respectively. Our main results are the explicit convergence rates of Hamming error, as a function of $(\vartheta, r, \rho)$, for different methods. We will study six methods: (i) Lasso as in (2); (ii) Elastic net (Zou & Hastie, 2005), which adds an additional $L^2$-penalty to (2), (iii) smoothly clipped absolute deviation (SCAD) (Fan & Li, 2001), which replaces the $L^1$-penalty by a non-convex penalty, (iv) thresholded Lasso (Zhou, 2009), which further thresholds the Lasso solution, and two iterative algorithms, (v) forward selection and (vi) forward backward selection (Huang et al., 2016); see Section 3 for a precise description of each method. To our best knowledge, our results are the first that directly compare Hamming errors of these methods.

## 2 A PREVIEW OF MAIN RESULTS AND SOME DISCUSSIONS

For any $\hat{\beta}$, its Hamming error is $H(\hat{\beta}, \beta) = \sum_{j=1}^{p} 1\{\hat{\beta}_j \neq 0, \beta_j = 0\} + \sum_{j=1}^{p} 1\{\hat{\beta}_j = 0, \beta_j \neq 0\}$. As we shall show, for any of the six methods studied here, there exists a function $h(\vartheta, r, \rho) \in [0, 1]$ such that $\mathbb{E}[H(\hat{\beta}, \beta)] = L_p p^{1-h(\vartheta, r, \rho)}$, where $L_p$ is a *multi*-$\log(p)$ term. (A multi-$\log(p)$ term is such that $L_p \cdot p^\epsilon \to \infty$ and $L_p \cdot p^{-\epsilon} \to 0$ for any $\epsilon > 0$.) Since the expected number of true relevant variables is $p^{1-\vartheta}$, we are interested in three cases:

- *Exact recovery*: $h(\vartheta, r, \rho) > 1$. In this case, the expected Hamming error is $o(1)$ as $p \to \infty$. It follows that model selection consistency holds.

- *Almost full recovery*: $\vartheta < h(\vartheta, r, \rho) < 1$. In this case, the expected Hamming error does not vanish as $p \to \infty$, but it is much smaller than the total number of true relevant variables. Variable selection is still satisfactory (although model selection consistency no longer holds).

- *No recovery*: $h(\vartheta, r, \rho) \leq \vartheta$. In this case, the expected Hamming error is comparable with or much larger than the total number of true relevant variables. Variable selection fails.

---

[1] In (1), we assume that each column of $X$ is standardized to have a unit $\ell^2$-norm and that the order of nonzero $\beta_j$ is $\sqrt{\log(n)}$. Alternatively, many works assume that each column of $X$ is standardized to have an $\ell^2$-norm of $\sqrt{n}$ and that the order for nonzero $\beta_j$ is $n^{-1/2}\sqrt{\log(p)}$. These are two equivalent parameterizations.

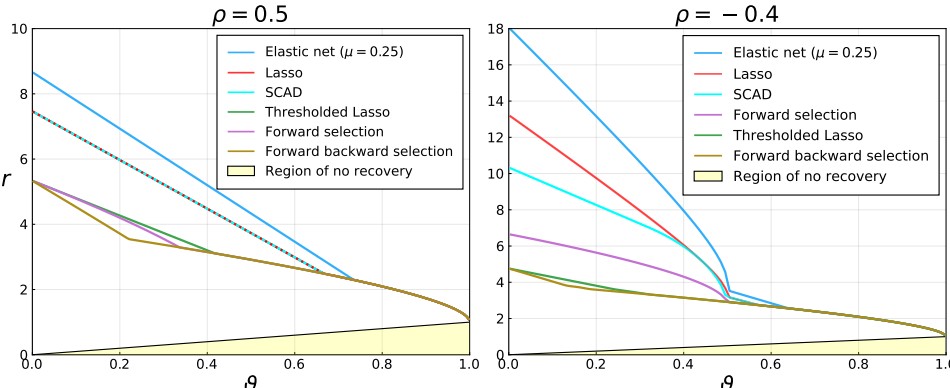

Figure 1: Phase diagrams of six variable selection methods for a block-wise diagonal design. The parameters $(\vartheta, r, \rho)$ characterize the sparsity, signal strength, and design correlations, respectively. For each method, we plot the curve $r = U(\vartheta)$ which separates Region of Almost Full Recovery and Region of Exact Recovery (the lower this curve, the better). Explicit forms of $U(\vartheta)$ are in Section 3. On the left panel, the curves for Lasso and SCAD overlap and are displayed as a dashed line. How to interpret these phase curves are discussed in Section 2.

We call the two-dimensional space $(\vartheta, r)$ the *phase space*. For each fixed $\rho$, the phase space is divided into three regions: *Region of Exact Recovery (ER)*, which is the subset $\{(\vartheta, r) : h(\vartheta, r, \rho) > 1\}$, and *Region of Almost Full Recovery (AFR)* and *Region of No Recovery (NR)* defined similarly. This gives rise to a *phase diagram* for each method. We denote the curve separating ER region and AFR region by $r = U(\vartheta)$ and the curve separating AFR region and NR region by $r = L(\vartheta)$; they are called the *upper* and *lower phase curves*, respectively. The phase diagram and phase curves are convenient ways to visualize the convergence rates of the Hamming error.

Figure 1 shows the phase curves for the six methods (with explicit expressions in the theorems in Section 3). These phase curves depend on the correlation parameter $\rho$. Under our model, for each diagonal block $(j, j+1)$, it holds that $\mathbb{E}[x_j' y | \beta] = \beta_j + \rho\beta_{j+1}$, where $\beta_j, \beta_{j+1} \in \{0, \tau_p\}$. Therefore, a *positive* $\rho$ boosts the signal at each individual site (i.e., $\mathbb{E}[x_j' y | \beta] \geq \beta_j$), while a *negative* $\rho$ leads to potential 'signal cancellation' (i.e., $\mathbb{E}[x_j' y | \beta] \leq \beta_j$). This is why the phase curves have different shapes for positive and negative $\rho$. In Figure 1, we plot the phase curves for $\rho = 0.5$ and $\rho = -0.4$. For other positive/negative value of $\rho$, the patterns are similar.

**Discussion of SCAD**. SCAD is a representative of non-convex penalization methods. There have been inspiring works that demonstrate the advantages of using a non-convex penalty (e.g., Fan & Peng (2004); Loh & Wainwright (2017)). Our results support their insights from a different angle: The phase curve of SCAD is strictly better than that of Lasso, when $\vartheta < 0.5$ and $\rho < 0$. Furthermore, our results illustrate where the advantage of SCAD comes from — compared with Lasso, it handles 'signal cancellation' better. To see this, we recall that 'signal cancellation' only happens for $\rho < 0$. Moreover, under our model (3), the expected number of signal pairs (a signal pair is a diagonal block $\{j, j+1\}$ where both $\beta_j$ and $\beta_{j+1}$ are nonzero) is $\asymp p\epsilon_p^2 = p^{1-2\vartheta}$. Therefore, 'signal cancellation' becomes problematic only when $\vartheta < 0.5$ and $\rho < 0$ both hold. This explains why the phase curves of SCAD and Lasso are the same for the other values of $\vartheta$ and $\rho$. We note that in the previous studies (e.g., Loh & Wainwright (2017)), the advantages of a non-convex penalty in handling 'signal cancellation' are reflected in the weaker conditions of $(X, \beta)$ for achieving model selection consistency. Our results support the advantage of using a non-convex penalty by directly studying the Hamming errors and phase diagrams.

The performance of SCAD can be further improved by adding an entry-wise thresholding on $\hat{\beta}$. We believe that the phase diagrams of *thresholded SCAD* are better than those of SCAD itself, although the extremely tedious analysis impedes us from specific results for now. Also, we are cautious about what to conclude from comparing SCAD and thresholded Lasso. In our settings, Lasso has no model selection consistency mainly because the signals are too weak (i.e., $r$ is not sufficiently large). In such settings, thresholded Lasso outperforms SCAD in terms of the Hamming error. However, there are cases where Lasso has no model selection consistency no matter how large the signal strength is

(Zhao & Yu, 2006). For those cases, it is possible that SCAD is better than thresholded Lasso (see Wainwright (2009) for a related study).

**Discussion of Elastic net**. The phase curve of Elastic net is worse than that of Lasso. As we will explain in Section 3.1, Elastic net is a 'bridge' between Lasso and marginal regression in our case. Since the phase curve of marginal regression is always worse than that of Lasso for the blockwise diagonal design, we do not benefit from using Elastic net in the current setting. We must note that Elastic net is motivated by genetic applications where several correlated variables are competing as predictors, and where it is implicitly assumed that groups of correlated variables tend to be all relevant or all irrelevant (Zou & Hastie, 2005). This is not captured by our model (3). Therefore, our results do not go against the benefits of Elastic net known in the literature, but rather our results support that the advantages of Elastic net come from 'group' appearance of signal variables.

**Discussion of thresholded Lasso**. Thresholded Lasso is a representative of improving Lasso by post-processing. There have been inspiring works that demonstrate the advantages of such a post-processing (van de Geer et al., 2011; Wang et al., 2020; Weinstein et al., 2020). Our results support these insights from a different angle. It is surprising (and very encouraging) that the improvement by post-Lasso thresholding is so significant. We note that Lasso is a 1-stage method, which solves a single optimization to obtain $\hat{\beta}$. By comparison, thresholded Lasso is a 2-stage method. Lasso has only one algorithm parameter $\lambda$, while thresholded Lasso has two algorithm parameters $\lambda$ and $t$ (the threshold). In Lasso, we control false positives and false negatives with the same algorithm parameter $\lambda$, and it is sometimes hard to find a value of $\lambda$ that simultaneously controls the two types of errors well. In thresholded Lasso, the two types of errors can be controlled separately by two algorithm parameters. This explains why thresholded Lasso enjoys such a big improvement upon Lasso. It inspires us to modify other 1-stage methods, such as SCAD, by adding a post-processing step of thresholding. For example, we conjecture that thresholded SCAD also has a strictly better phase diagram than that of Lasso, even for a positive $\rho$. On the other hand, thresholding is no free lunch. It leaves one more tuning parameter to be decided in practice. Our theoretical results are based on ideal tuning parameters. How to properly choose these tuning parameters in a data-driven way is an interesting question. Weinstein et al. (2020) proposes a promising approach, where they use cross-validation to select $\lambda$ and FDR control by knockoff to select $t$. We leave it for future work to study the phase diagrams with data-driven tuning parameters.

**Discussion of the two iterative algorithms**. We consider two iterative algorithms, forward selection ('Forward') and forward backward selection ('FB'). The FB algorithm we analyze is a simplified version in Huang et al. (2016), which has only one backward step (after all the forward steps have finished) by thresholding the refitted least-squares solution. Our results show that both methods outperform Lasso, and between these two methods, FB is strictly better than Forward. In the literature, there are very interesting theoretical works showing the advantages of iterative algorithms for variable selection (Donoho et al., 2012; Zhang, 2011). Our results support their insights from a different angle. We discover that, for a wide range of $\rho$, FB has the best phase diagram among all the six methods. This is a very encouraging result. Of course, it is as important to note that the performance of an iterative algorithm tends to be more sensitive to the form of the design, due to its sequential nature.

## 3 MAIN RESULTS

Consider model (1), (3), and (4), where we set $\sigma^2 = 1$ without loss of generality. Let $\mathbb{E}[H(\hat{\beta}, \beta)]$ be the expected Hamming error, where the expectation is with respect to the randomness of $\beta$ and $z$. Let $L_p$ denote a generic multi-log($p$) term such that $L_p p^\epsilon \to \infty$ and $L_p p^{-\epsilon} \to 0$ for any $\epsilon > 0$.

**Theorem 1.** *Under Models* (1)*,* (3)*, and* (4)*, for each of the methods considered in this paper (Lasso, Elastic net, SCAD, thresholded Lasso, forward selection, forward backward selection, as well as marginal regression in Section 3.1), there exists a function $h(\vartheta, r, \rho)$ such that $\mathbb{E}[H(\hat{\beta}, \beta)] = L_p p^{1-h(\vartheta, r, \rho)}$. The explicit expressions of $h(\vartheta, r, \rho)$, which may depend on the tuning parameters of a method, are given in Theorems B.1, C.1, D.1-D.3, F.1, G.1, H.1-H.4 of the supplement.*

In the main article, to save space, we only present the expressions of the upper phase curve $U(\vartheta) = U(\vartheta; \rho)$ and the lower phase curve $L(\vartheta) = L(\vartheta; \rho)$ for each method, which are defined as follows:

$$U(\vartheta; \rho) = \inf\{r > 0 : h(\vartheta, r, \rho) > 1\}, \qquad L(\vartheta; \rho) = \inf\{r > 0 : h(\vartheta, r, \rho) > \vartheta\}. \tag{5}$$

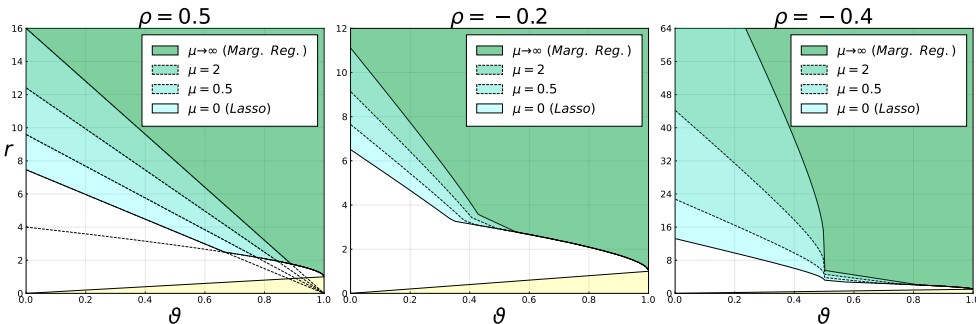

Figure 2: The phase diagrams of Elastic net and its comparison with Lasso (notation: $\eta = \rho/(1+\mu)$).

These two curves describe the phase diagram: The upper phase curve $U(\vartheta)$ separates the ER region and AFR region, and the lower phase curve $L(\vartheta)$ separates the AFR region and NR region.

### 3.1 ELASTIC NET AND LASSO

The Elastic net (Zou & Hastie, 2005) is a method that estimates $\beta$ by

$$\hat{\beta}^{\text{EN}} = \text{argmin}_\beta \{\|y - X\beta\|^2/2 + \lambda\|\beta\|_1 + (\mu/2)\|\beta\|^2\}. \tag{6}$$

Compared with Lasso, it adds an additional $L^2$-penalty to the objective function. Below, we fix $\mu > 0$ and re-parametrize $\lambda = \sqrt{2q\log(p)}$, for some constant $q > 0$. The choice of $q$ affects the exponent, $1 - h(\vartheta, r, \rho)$, in the expression of $\mathbb{E}[H(\hat{\beta}, \beta)]$. We choose the ideal $q$ that minimizes $1 - h(\vartheta, r, \rho)$. The next theorem is proved in the supplement.

**Theorem 2** (Elastic Net). *Under Models* (1), (3), *and* (4), *let* $\hat{\beta}^{\text{EN}}$ *be the Elastic net estimator in* (6). *Fix* $\mu$ *and write* $\eta = \rho/(1+\mu)$. *Let* $\lambda = \sqrt{2q\log(p)}$ *with an ideal choice of* $q$ *that minimizes the exponent of* $\mathbb{E}[H(\hat{\beta}, \beta)]$. *The phase curves are given by* $L(\vartheta) = \vartheta$, *and*

$$U(\vartheta) = \begin{cases} \max\{h_1(\vartheta), h_2(\vartheta)\}, & \text{when } \rho \geq 0, \\ \max\{h_1(\vartheta), h_2(\vartheta), h_3(\vartheta), h_4(\vartheta)\}, & \text{when } \rho < 0, \end{cases}$$

*where* $h_1(\vartheta) = (1 + \sqrt{1-\vartheta})^2$, $h_2(\vartheta) = \left(\frac{1-|\eta|}{1-|\rho|} + \frac{\sqrt{1+\eta^2-2\rho\eta}}{1-|\rho|}\right)^2(1-\vartheta)$, $h_3(\vartheta) = \frac{1}{(1-|\rho|)^2}\left(1 + \frac{\sqrt{1+\eta^2-2\rho\eta}}{1+|\eta|}\sqrt{1-2\vartheta}\right)^2$, *and* $h_4(\vartheta) = \frac{1+\eta^2-2\rho\eta}{(1-2|\rho|+\rho\eta)_+^2}\left(\sqrt{1-\vartheta} + \frac{1-|\eta|}{1+|\eta|}\sqrt{1-2\vartheta}\right)^2$.

Lasso is a special case with $\mu = 0$. By setting $\mu = 0$ (equivalently, $\eta = \rho$) in Theorem 2, we obtain the phase curves for Lasso. They agree with the results in Ji & Jin (2012) (but Ji & Jin (2012) does not cover Elastic net).

To see the effect of the $L^2$-penalty, we consider an extreme case where $\mu \to \infty$. Some elementary algebra shows that $(1+\mu)\hat{\beta}^{\text{EN}}$ converges to the soft-thresholding of $X'y$ at the threshold $\lambda$. In other words, as $\mu \to \infty$, Elastic net converges to marginal screening (i.e., select variables by thresholding the marginal regression coefficients). At the same time, when $\mu = 0$, $(1+\mu)\hat{\beta}^{\text{EN}}$ equals the Lasso estimate. Hence, Elastic net serves as a bridge between Lasso and marginal regression. In the setting here, the phase diagram of marginal regression is inferior to that of Lasso, and so the phase diagram of Elastic net is also inferior to that of Lasso. See the proposition below and Figure 2:

**Proposition 1.** *In Theorem 2, for each fixed* $\vartheta \in (0,1)$, *as* $\mu \to 0$, $U(\vartheta)$ *is monotone decreasing and converges to* $U_{\text{Lasso}}(\vartheta)$, *which is the upper phase curve of Lasso; as* $\mu \to \infty$, $U(\vartheta)$ *is monotone increasing and converges to* $U_{\text{MR}}(\vartheta)$, *which is the upper phase curve of marginal regression. Furthermore, when* $\rho \leq -\frac{1}{2}$, $U_{\text{MR}}(\theta) = \infty$ *for all* $0 < \vartheta \leq \frac{1}{2}$ *(i.e. exact recovery is impossible to achieve no matter how large* $r$ *is).*

### 3.2 SMOOTHLY CLIPPED ABSOLUTE DEVIATION PENALTY (SCAD)

SCAD (Fan & Li, 2001) is a non-convex penalization method. For any $a > 2$, it defines a penalty function $q_\lambda(\theta)$ on $(0,\infty)$ by $q_\lambda(\theta) = \int_0^\theta q'_\lambda(t)dt$, where $q'_\lambda(\theta) = \lambda\{I(\theta \leq \lambda) + \frac{(a\lambda-\theta)_+}{(a-1)\lambda}I(\theta > \lambda)\}$.

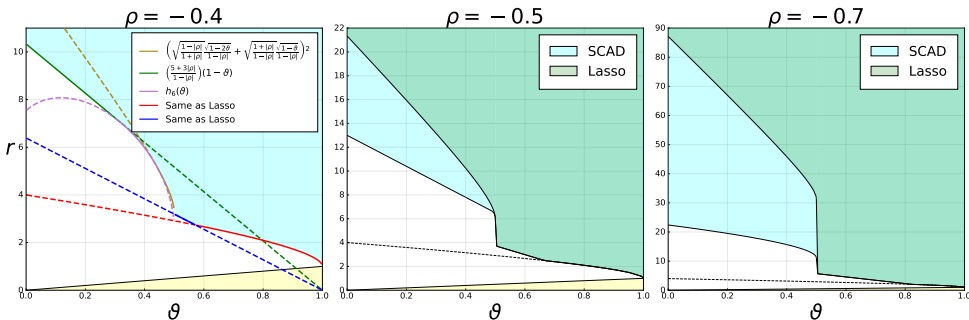

Figure 3: Left: Phase curves of SCAD. Middle and Right: Comparison of SCAD and Lasso.

The resulting penalty function $q_\lambda(\cdot)$ coincides with the $L^1$-penalty in $(0, \lambda]$ and becomes a constant in $[a\lambda, \infty)$. Let $Q_\lambda(\beta) = \sum_{j=1}^p q_\lambda(|\beta_j|)$. Then, SCAD estimates $\beta$ by

$$\hat{\beta}^{\mathrm{SCAD}} = \mathrm{argmin}_\beta \{\|y - X\beta\|^2/2 + Q_\lambda(\beta)\}. \tag{7}$$

The following theorem is proved in the supplemental material (see Figure 3, left panel):

**Theorem 3** (SCAD). *Under Models* (1), (3), *and* (4), *let* $\hat{\beta}^{\mathrm{SCAD}}$ *be the SCAD estimator in* (7). *Fix* $a \in (2, \frac{2}{1-|\rho|})$. *Let* $\lambda = \sqrt{2q\log(p)}$ *with an ideal choice of q that minimizes the rates of convergence of the expected Hamming error. The phase curves are given by* $L(\vartheta) = \vartheta$, *and*

$$U(\vartheta) = \begin{cases} \max\{h_1(\vartheta), h_2(\vartheta), h_3(\vartheta)\}, & when \ \rho \geq 0, \\ \max\{h_1(\vartheta), h_2(\vartheta), h_4(\vartheta), h_5(\vartheta)\}, & when \ \rho < 0, \end{cases}$$

*where* $h_1(\vartheta) = (1 + \sqrt{1-\vartheta})^2$, *and* $h_2(\vartheta) = \left(1 + \sqrt{\frac{1+|\rho|}{1-|\rho|}}\right)^2(1-\vartheta)$, $h_4(\vartheta) = \left(\sqrt{\frac{1-2\vartheta}{1-|\rho|^2}} + \frac{1}{1-|\rho|}\right)^2$,

$h_3(\vartheta) = \left(\frac{3+\rho}{2(1-\rho^2)}\sqrt{\frac{1+\rho}{1-\rho}}\sqrt{1-\vartheta} + \frac{1}{2}\sqrt{\frac{2(1-2\vartheta)}{1+\rho} - \frac{(1-\vartheta)}{(1-\rho)^2}}\right)^2$, *and*

$$h_5(\vartheta) = \begin{cases} \left(\frac{5+3|\rho|}{1-|\rho|}\right)(1-\vartheta), & if \ \sqrt{\frac{1-2\vartheta}{1-\vartheta}} \geq \frac{3-4|\rho|-3\rho^2}{(1-|\rho|)}\sqrt{\frac{1+|\rho|}{5+3|\rho|}}, \\ \frac{1}{(1-|\rho|)^2}\left(\sqrt{\frac{1+|\rho|}{1-|\rho|}}\sqrt{1-\vartheta} + \sqrt{\frac{1-|\rho|}{1+|\rho|}}\sqrt{1-2\vartheta}\right)^2, & if \ \sqrt{\frac{1-2\vartheta}{1-\vartheta}} \leq \frac{(1+|\rho|)(1-2|\rho|)}{1-|\rho|}, \\ h_6(\vartheta) & other \ wise, \end{cases}$$

*with*

$$h_6(\vartheta) = \left\{\sqrt{\frac{1-2\vartheta}{1-\rho^2}} + \frac{\frac{1-2|\rho|}{1-|\rho|}\sqrt{\frac{1-2\vartheta}{1-\rho^2}} + \sqrt{\left[\left(\frac{1-2|\rho|}{1-|\rho|}\right)^2 + \frac{1-|\rho|}{1+|\rho|}\right](1-\vartheta) - \frac{1-2\vartheta}{(1+|\rho|)^2}}}{(1-|\rho|)\left[\left(\frac{1-2|\rho|}{1-|\rho|}\right)^2 + \frac{1-|\rho|}{1+|\rho|}\right]}\right\}^2.$$

Note that the phase curves of Lasso are given in Theorem 2 by setting $\eta = \rho$. We compare SCAD with Lasso. When $\rho < 0$, the upper phase curve in Theorem 3 is strictly lower than that of Lasso (see Figure 3, middle and right panels). When $\rho \geq 0$, the upper phase curve in Theorem 3 is sometimes higher than that of Lasso. Note that we restrict $a < \frac{2}{1-|\rho|}$ in Theorem 3. In fact, a larger $a$ may be preferred for $\rho \geq 0$. The next proposition is about using an optimal $a$.

**Proposition 2.** *In the SCAD estimator, we choose* $a = a^*$ *and* $\lambda = \sqrt{2q^* \log(p)}$ *such that* $(a^*, q^*) = (a^*(\vartheta, r, \rho), q^*(\vartheta, r, \rho))$ *minimize the rates of convergence of the expected Hamming error among all choices of* $(a, q)$. *Let* $U^*(\vartheta)$ *be the resulting upper phase curve for SCAD. Then,* $U(\vartheta) = U_{\mathrm{Lasso}}(\vartheta)$ *when* $\rho \geq 0$, *and* $U(\vartheta) < U_{\mathrm{Lasso}}(\vartheta)$ *when* $\rho < 0$.

The phase curves of SCAD are insensitive to the choice of $a$. When $a < 0$, the optimal $a^*$ can be any value in $(2, \frac{2}{1-|\rho|})$. When $\rho \geq 0$, there exists a constant $c = c(\vartheta, \rho)$ such that the optimal $a^*$ is any value in $(c, \infty)$. As $a \to \infty$, the SCAD penalty reduces to the $L^1$-penalty. This explains why the phase curve of SCAD is the same as that of Lasso when $\rho \geq 0$.

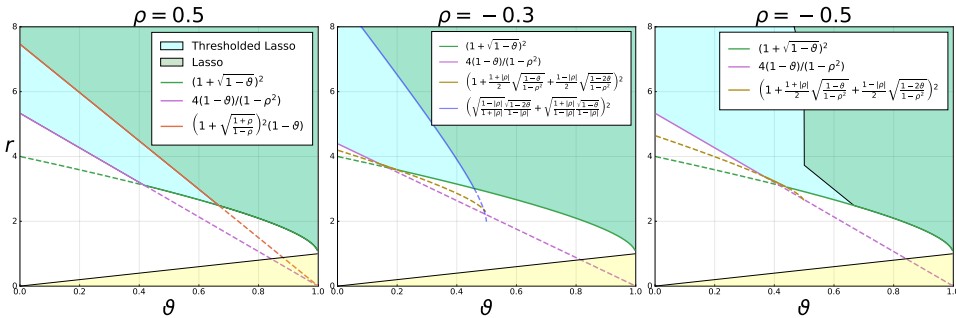

Figure 4: Comparison of the phase diagrams of thresholded Lasso and Lasso.

## 3.3 THRESHOLDED LASSO

Let $\hat{\beta}^{\mathrm{Lasso}}$ be the Lasso estimator in (2). The thresholded Lasso estimator $\hat{\beta}^{\mathrm{TL}}$ is obtained by applying coordinate-wise hard-thresholding to the Lasso estimator:

$$\hat{\beta}_j^{\mathrm{TL}} = \hat{\beta}^{\mathrm{Lasso}} \cdot 1\{|\hat{\beta}^{\mathrm{Lasso}}| > t\}, \qquad 1 \leq j \leq p. \tag{8}$$

**Theorem 4** (Thresholded Lasso). *Under Models* (1), (3), *and* (4), *let* $\hat{\beta}^{\mathrm{TL}}$ *be the thresholded Lasso estimator in* (8). *Let* $\lambda = \sqrt{2q\log(p)}$ *and* $t = \sqrt{2w\log(p)}$ *with the ideal* $(q, w)$ *that minimize the exponent of the expected Hamming error. The phase curves are given by* $L(\vartheta) = \vartheta$, *and*

$$U(\vartheta) = \begin{cases} \max\{h_1(\vartheta), h_2(\vartheta)\}, & \text{when } \rho \geq 0, \\ \max\{h_1(\vartheta), h_2(\vartheta), h_3(\vartheta)\}, & \text{when } \rho < 0, \end{cases}$$

*where* $h_1(\vartheta) = (1 + \sqrt{1-\vartheta})^2$, $h_2(\vartheta) = \frac{4(1-\vartheta)}{1-\rho^2}$, *and* $h_3(\vartheta) = \left(1 + \frac{1+|\rho|}{2}\sqrt{\frac{1-\vartheta}{1-\rho^2}} + \frac{1-|\rho|}{2}\sqrt{\frac{1-2\vartheta}{1-\rho^2}}\right)^2$.

See Figure 4 for a comparison with Lasso (a special case of $t = 0$). With the flexibility of using an optimal $t$, the phase diagram of thresholded Lasso is always better than that of Lasso.

Theorem 4 also gives other interesting facts about thresholded Lasso. First, the shape of phase curves is much less affected by the sign of $\rho$. This differs from Lasso, Elastic net, and SCAD, for which the shape of phase curves is significantly different for positive and negative $\rho$. Second, the optimal $\lambda$ in thresholded Lasso is considerably smaller than the optimal $\lambda$ in Lasso (it can be seen from the proofs of Theorem 4 and Theorem 2). This is because the $\lambda$ in thresholded Lasso only serves to control false negatives, but the $\lambda$ in Lasso is used to simultaneously control false positives and false negatives, hence, cannot be too small. We observe the same phenomenon in simulations; see Section 4.

## 3.4 FORWARD SELECTION AND FORWARD BACKWARD SELECTION

Forward selection is a classical textbook method for variable selection. Write $X = [x_1, x_2, \ldots, x_p]$, where $x_i \in \mathbb{R}^n$ for $1 \leq i \leq p$. For any subset $A \subset \{1, 2, \ldots, p\}$, let $P_A^\perp$ be the projection onto the orthogonal complement of the linear space spanned by $\{x_i : i \in A\}$. Given a threshold $t > 0$, the forward selection algorithm initializes with $S_0 = \emptyset$ and $\hat{r}_0 = y$. At the $k$th iteration, compute

$$i^* = \mathrm{argmax}_{i \notin S_{k-1}} |x_i' \hat{r}_{k-1}|, \qquad \delta = |x_{i^*}' \hat{r}_{k-1}| / \|P_{S_{k-1}}^\perp x_{i^*}\|.$$

If $\delta > t$, compute $S_k = S_{k-1} \cup \{i^*\}$ and $\hat{r}_k = P_{S_k}^\perp y$; otherwise, output $\hat{\beta}^{\mathrm{forward}}$ as the least-squares estimator restricted to $S_{k-1}$. The stopping rule of $\delta \leq t$ is equivalent to measuring the decrease of the residual sum of squares. The following theorem is proved in the supplemental material:

**Theorem 5** (Forward Selection). *Under Models* (1), (3), *and* (4), *let* $\hat{\beta}^{\mathrm{forward}}$ *be the estimator from forward selection. Let* $t = \sqrt{2q\log(p)}$ *with the ideal* $q$ *that minimizes the exponent of the expected Hamming error. The phase curves are given by* $L(\vartheta) = \vartheta$, *and*

$$U(\vartheta) = \begin{cases} \max\{h_1(\vartheta), h_2(\vartheta), h_3(\vartheta)\}, & \text{when } \rho \geq 0, \\ \max\{h_1(\vartheta), h_2(\vartheta), h_3(\vartheta), h_4(\vartheta)\}, & \text{when } \rho < 0, \end{cases}$$

*with* $h_1(\vartheta) = (1 + \sqrt{1-\vartheta})^2$, $h_2(\vartheta) = \frac{2(1-\vartheta)}{1-|\rho|}$, $h_3(\vartheta) = \frac{(1+\sqrt{1-2\vartheta})^2}{1-\rho^2}$, $h_4(\vartheta) = \left(\sqrt{\frac{1-2\vartheta}{2(1-|\rho|)}} + \frac{1}{1-|\rho|}\right)^2$.

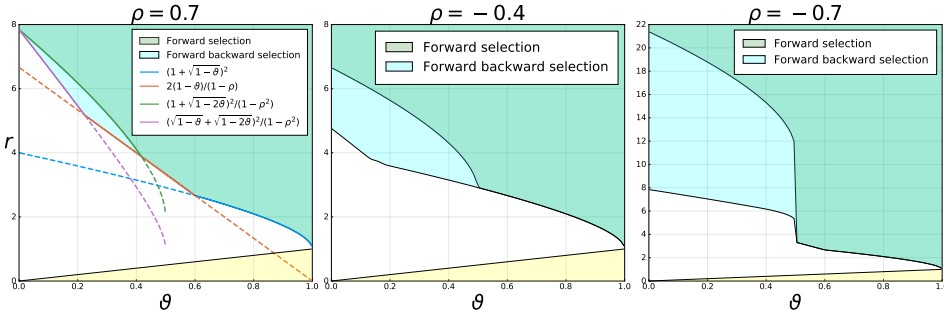

Figure 5: The phase diagrams of forward selection and forward backward selection.

Forward backward selection (FB) modifies forward selection by allowing to drop variables. We use the FB algorithm in Huang et al. (2016), where the backward step is conducted after all the forward steps are finished. For a threshold $v > 0$, it applies entry-wise thresholding on $\hat{\beta}^{\text{forward}}$:

$$\hat{\beta}_j^{\text{FB}} = \hat{\beta}_j^{\text{forward}} \cdot 1\{|\hat{\beta}_j^{\text{forward}}| > v\}, \qquad 1 \leq j \leq p. \tag{9}$$

**Theorem 6** (Forward Backward Selection). *Under Models* (1), (3), *and* (4), *let* $\hat{\beta}^{\text{FB}}$ *be the estimator from forward selection. Let* $t = \sqrt{2q\log(p)}$ *and* $v = \sqrt{2u\log(p)}$ *with the ideal* $(q, u)$ *that minimize the exponent of the expected Hamming error. When* $\rho \geq 0$, *the phase curves are given by* $L(\vartheta) = \vartheta$, *and*

$$U(\vartheta) = \max\left\{h_1(\vartheta), h_2(\vartheta), h_3^*(\vartheta)\right\},$$

*where* $h_1(\vartheta)$ *and* $h_2(\vartheta)$ *are the same as in Theorem 5 and* $h_3^*(\vartheta) = \frac{(\sqrt{1-\vartheta}+\sqrt{1-2\vartheta})^2}{1-\rho^2}$. *When* $\rho < 0$,

$$U(\vartheta) \leq \max\left\{g_1(\vartheta), g_2(\vartheta), g_3(\vartheta), g_4(\vartheta)\right\},$$

*where* $g_1(\vartheta) = (v_{\min}(\vartheta) + \sqrt{1-\vartheta})^2$, $g_2(\vartheta) = \frac{2(1-\vartheta)}{1-|\rho|}$, $g_3(\vartheta) = \left(\sqrt{\frac{1-2\vartheta}{1-\rho^2}} + v_{\min}(\vartheta)\right)^2$, $g_4(\vartheta) = \left(\sqrt{\frac{1-2\vartheta}{2(1-|\rho|)}} + \frac{t_{\min}(\vartheta)}{1-|\rho|}\right)^2$, $v_{\min}(\vartheta) = \max\{1, \sqrt{\frac{1-\vartheta}{1-\rho^2}}\}$, *and* $t_{\min}(\vartheta) = \max\{\frac{\sqrt{2}}{2}, \frac{v_{\min}(\vartheta)}{1+|\rho|/\sqrt{1-\rho^2}}\}$.

Theorem 6 gives $U(\vartheta)$ for $\rho \geq 0$ and an upper bound of it for $\rho < 0$. Combining it with Theorems 2 and 5, we conclude that the upper phase curve of FB is always better than those of Lasso and forward selection (for $\rho < 0$, the upper bound here is already better than $U(\vartheta)$ for the other two methods).

We remark that we did obtain the exact phase curve for $\rho < 0$ in the proof of Theorem 6. It is just too complicated and space-consuming to present it in the main text. However, given specific values of $(\vartheta, \rho)$, we can always plot the exact phase curve using the (complicated) formulas in the supplement. In Figures 1 and 5, the phase curves of FB are indeed the exact ones.

### 3.5 CONNECTION TO THE RANDOM DESIGN MODEL

Consider the random design as mentioned in Section 1. The *minimax Hamming error* is $H^*(\vartheta, r, \rho) = \inf_{\hat{\beta}} \mathbb{E}[H(\hat{\beta}, \beta)]$, where the infimum is taken over all methods $\hat{\beta}$ and the expectation is with respect to the randomness of $(X, \beta, z)$. We can define $H^*(\vartheta, r, \rho)$ in the same way for our current model (4). The minimax Hamming error is related to the statistical limit of the model setting, but not any specific method. The next theorem shows that, when $n \gg s_p = p^{1-\vartheta}$ (we allow both $p \leq n$ and $p > n$), the convergence rate of the minimax Hamming error is the same under two models.

**Theorem 7.** *Under Models* (1) *and* (3), *suppose* $X$ *is independent of* $(\beta, z)$ *and its rows are iid generated from* $\mathcal{N}(0, n^{-1}\Sigma)$, *with* $\Sigma$ *having the same form as* $G$ *in* (4). *Suppose* $n = p^\omega$, *with* $\omega > 1 - \vartheta$ *(note: this allows* $\omega < 1$, *which corresponds to* $n \ll p$). *There exists a number* $h^{**}(\vartheta, r, \rho)$ *such that the minimax Hamming error satisfies that* $H^*(\vartheta, r, \rho) = L_p p^{1-h^{**}(\vartheta, r, \rho)}$. *Furthermore, if we instead have* $X'X = \Sigma$ *(i.e., model* (4)), *then it also holds that* $H^*(\vartheta, r, \rho) = L_p p^{1-h^{**}(\vartheta, r, \rho)}$.

## 4 SIMULATIONS

In Experiments 1-3, $(n, p) = (1000, 300)$. In Experiment 4, $(n, p) = (500, 1000)$.

**Experiment 1** (block-wise diagonal designs). We generate $(X, \beta)$ as in (3)-(4). For each method, we select the ideal tuning parameters that minimize the average Hamming error over 50 repetitions. The averaged Hamming errors and its standard deviations under the ideal tuning parameters over 500 repetitions are reported below. The results are consistent with the theoretical phase diagrams (see Figure 1). E.g., thresholded Lasso and forward backward selection are the two methods that perform the best; Lasso is more unsatisfactory when $\rho < 0$; SCAD improves Lasso when $\rho < 0$.

| $\rho$ | $\vartheta$ | $r$ | Lasso | ThresLasso | ElasticNet | SCAD | Forward | FoBackward |
|---|---|---|---|---|---|---|---|---|
| 0.5 | 0.1 | 1.5 | 11.57 (3.59) | **10.48** (3.34) | 11.57 (3.31) | 11.72 (3.33) | 14.88 (4.12) | 13.35 (3.90) |
| 0.5 | 0.1 | 4 | 1.00 (1.00) | **0.42** (0.65) | 1.03 (1.00) | 1.00 (0.96) | 0.66 (0.84) | 0.51 (0.73) |
| -0.5 | 0.1 | 1.5 | 35.62 (5.09) | 15.62 (4.06) | 35.48 (5.64) | 25.87 (5.04) | 19.48 (4.61) | **14.82** (3.82) |

Table 1: Experiment 1 (block-diagonal designs). $(n, p) = (1000, 300)$.

**Experiment 2** (general designs). In the Toeplitz design, we let $(X'X)_{i,j} = 0.7^{|i-j|}$ and set $(\vartheta, r) = (0.1, 2.5)$. In the factor model design, we let $X'X = BB' - \text{diag}(BB') + I_p$, where entries of $B \in \mathbb{R}^{p \times 2}$ are *iid* from $\text{Unif}(0, 0.6)$, and set $(\vartheta, r) = (0.1, 1.5)$. Same as in Experiment 1, we use the ideal tuning parameters. The averaged Hamming errors and its standard deviations are reported below. The Toeplitz design is a setting where each variable is only highly correlated with a few other variables. The factor model design is a setting where a variable is (weakly) correlated with all the other variables. The results are quite similar to those in Experiment 1. This confirms that the insight gained in the study of the block-wise diagonal design continues to apply to more general designs.

| design | Lasso | ThresLasso | ElasticNet | SCAD | Forward | FoBackward |
|---|---|---|---|---|---|---|
| Toeplitz | 47.15 (6.32) | **22.02** (5.31) | 47.40 (6.41) | 24.61 (5.70) | 30.77 (6.18) | 22.93 (5.44) |
| Factor model | 21.14 (4.52) | **15.90** (3.87) | 21.20 (4.45) | 19.68 (4.23) | 20.04 (4.34) | 16.13 (3.76) |

Table 2: Experiment 2 (general designs). $(n, p) = (1000, 300)$.

**Experiment 3** (tuning parameters). Fix $(\vartheta, r) = (0.1, 1.5)$ and $\rho \in \{\pm 0.5\}$ in the block-wise diagonal design. We study the effect of tuning parameters in Lasso, thresholded Lasso (ThreshLasso), forward selection (ForwardSelect), and forward backward selection (FB). In (a)-(b), we show the heatmap of averaged Hamming error (over 50 repetitions) of ThreshLasso for a grid of $(t, \lambda)$; when $t = 0$, it reduces to Lasso. In (c)-(d), we show the Hamming error of FB for a grid of $(v, t)$; when $v = 0$, it reduces to ForwardSelect. Cyan points are theoretically optimal tuning parameters (formulas are in proofs of theorems). Red points are empirically optimal tuning parameters that minimize the averaged Hamming error. The theoretical tuning parameter values are quite close to the empirically optimal ones. Moreover, the optimal $\lambda$ in ThreshLasso is smaller than the optimal $\lambda$ in Lasso.

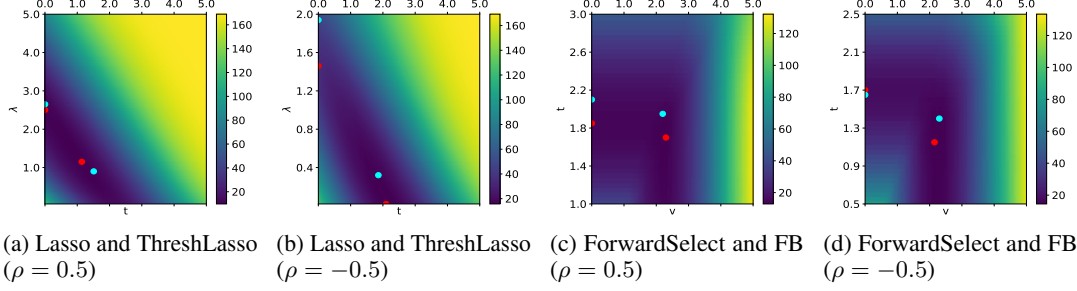

(a) Lasso and ThreshLasso ($\rho = 0.5$)  (b) Lasso and ThreshLasso ($\rho = -0.5$)  (c) ForwardSelect and FB ($\rho = 0.5$)  (d) ForwardSelect and FB ($\rho = -0.5$)

Figure 6: Experiment 3 (effects of tuning parameters). In all plots, cyan points are computed from the formulas in our theory, and red points are the empirically best tuning parameters (they minimize the average Hamming error over 500 repetitions). In (a)-(b), the cyan/red points with $t = 0$ correspond to Lasso, and the other two are for thresholded Lasso. In (c)-(d), the cyan/red points with $t = 0$ correspond to forward selection, and the other two are for forward backward selection.

**Experiment 4** ($p > n$ and random designs). Fix $(n, p, \vartheta, r) = (500, 1000, 0.5, 1.5)$. We simulate data from the random design setting in Theorem 7. We study the average Hamming error over 500 repetitions (tuning parameters are set in the same way as in Experiment 1). See Table 3. We have some similar observations as before: e.g., ThreshLasso and FoBackward are still the best two,

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

| $\rho$ | $\vartheta$ | $r$ | Lasso | ThresLasso | ElasticNet | SCAD | Forward | FoBackward |
|---|---|---|---|---|---|---|---|---|
| 0.5 | 0.5 | 1.5 | 16.02 (5.52) | **9.83** (4.08) | 13.92 (5.12) | 15.98 (6.28) | 11.74 (5.55) | 9.84 (4.93) |
| -0.5 | 0.5 | 1.5 | 18.49 (6.03) | 10.50 (4.23) | 15.18 (5.64) | 18.12 (6.00) | 12.00 (5.67) | **10.41** (5.03) |

Table 3: Experiment 4 ($p > n$ and random designs).

Xiaotong Shen, Wei Pan, and Yunzhang Zhu. Likelihood-based selection and sharp parameter estimation. *Journal of the American Statistical Association*, 107(497):223–232, 2012.

Wenguang Sun and T Tony Cai. Oracle and adaptive compound decision rules for false discovery rate control. *Journal of the American Statistical Association*, 102(479):901–912, 2007.

Robert Tibshirani. Regression shrinkage and selection via the lasso. *Journal of the Royal Statistical Society: Series B (Statistical Methodology)*, 58(1):267–288, 1996.

Sara van de Geer, Peter Bühlmann, and Shuheng Zhou. The adaptive and the thresholded lasso for potentially misspecified models (and a lower bound for the lasso). *Electronic Journal of Statistics*, 5:688–749, 2011.

Martin J Wainwright. Sharp thresholds for high-dimensional and noisy sparsity recovery using $L_1$-constrained quadratic programming (lasso). *IEEE Transactions on Information Theory*, 55(5): 2183–2202, 2009.

Shuaiwen Wang, Haolei Weng, and Arian Maleki. Which bridge estimator is optimal for variable selection? *The Annals of Statistics*, 48(5), 2020.

Asaf Weinstein, Weijie Su, Małgorzata Bogdan, Rina F Barber, and Emmanuel J Candès. A power analysis for knockoffs with the lasso coefficient-difference statistic. *arXiv preprint arXiv:2007.15346*, 2020.

Cun-Hui Zhang. Nearly unbiased variable selection under minimax concave penalty. *The Annals of Statistics*, 38(2):894–942, 2010.

Tong Zhang. Adaptive forward-backward greedy algorithm for learning sparse representations. *IEEE Transactions on Information Theory*, 57(7):4689–4708, 2011.

Peng Zhao and Bin Yu. On model selection consistency of lasso. *The Journal of Machine Learning Research*, 7:2541–2563, 2006.

Shuheng Zhou. Thresholding procedures for high dimensional variable selection and statistical estimation. *Advances in Neural Information Processing Systems*, 22:2304–2312, 2009.

Hui Zou. The adaptive lasso and its oracle properties. *Journal of the American Statistical Association*, 101(476):1418–1429, 2006.

Hui Zou and Trevor Hastie. Regularization and variable selection via the elastic net. *Journal of the Royal Statistical Society: Series B (Statistical Methodology)*, 67(2):301–320, 2005.

