# OpenReview forum: "A Comparison of Hamming Errors of Representative Variable Selection Methods"
_ICLR.cc/2022/Conference — ICLR 2022 Poster_

### Official Review · Reviewer_fogT · 2021-10-28

**Correctness:** 4
**Technical Novelty And Significance:** 3
**Empirical Novelty And Significance:** 3
**Recommendation:** 6
**Confidence:** 3

**Main Review:**

## Strengths
The paper is well written and easy to follow. Since variable selection is of importance to many, the problem is well motivated and worthy of study. The phase diagrams and insights provided in the discussion are useful for developing intuition about each method and laying a foundation for how future work might be carried out. Although I have not reviewed proofs closely, they appear to be of sufficient technical merit and illustrate the authors mettle!

## Weaknesses
The primary weakness of the paper seems to be in the applicability to realistic problems. There is brief mention in section 2 of how the block structure can be found in the correlation matrix of genetic markers and how empirical results for Toeplitz matrices agree with the simplified theory, but little else is said. A few words motivating the use of such a simplified model with additional examples would be useful and make the paper more compelling.


## Questions and comments:
- Does $X^T X = I_p \ne \text{diag}(X^T X) = [1,1, \dots, 1]^T$ in equation (1)? The notation is a bit unclear.
- Consider writing "smoothly clipped absolute deviation" before using the acronym.
- A few words defining the multi-log function would be helpful for a general audience.



**Summary Of The Paper:**

This paper investigates the theoretical performance of a number of variables selection methods when the columns of the design matrix are correlated. They focus specifically on the case where the Gram matrix for the design is block-wise diagonal (2 x 2 blocks) with the off-diagonals within each block given by a correlation parameter and diagonals set to 1. The quality of each estimator is evaluated by using its expected Hamming distance. Extending ideas presented in _UPS delivers optimal phase diagram in high-dimensional variable selection [Ji and Jin]_ and others, phase diagrams are constructed with phase curves (derived in the appendices) establishing the boundary between each region: exact recovery, almost full recovery, and no recovery. A discussion of each variable selection method provides insight and a comparison between them is included.

**Summary Of The Review:**

Given the complexity of calculations as mentioned in the _technical novelty_ section, it isn't clear that more realistic settings can be addressed, i.e., larger blocks, more complicated correlation models, etc., with analytic phase curves. 71 pages of proofs suggest that small increases in model complexity will quickly result in prohibitively complex calculations. Although I believe the analysis of this idealized model is of sufficient interest to researchers in variable selection methods, it would be nice to know that the results can reasonably be used by a wider audience.

I would like to know that the simplified block correlated model is in fact a reasonable approximation for a number of real examples or that phase diagrams for more realistic models are within striking distance. For now, I'd say the paper is marginally below the acceptance threshold and ask for the authors to further motivate the simplified model.


====== REVIEW UPDATE ========
Based on the authors comments, I am improving my recommendation from a 5 to a 6 provided the authors incorporate their response into the paper (I still haven't seen a revision).

---

### Official Review · Reviewer_3QXZ · 2021-10-29

**Correctness:** 3
**Technical Novelty And Significance:** 3
**Empirical Novelty And Significance:** 3
**Recommendation:** 6
**Confidence:** 3

**Main Review:**

Basically, I like the paper's motivation; it is quite a fundamental research problem to select unnecessary variables from a given data matrix. Besides, this paper is well structured, and the theoretical background of the proposed approach is well described in the paper. Specifically, I really appreciate that this paper theoretically revealed Hamming errors of each variable selection approach.

Unfortunately, the theoretical analyses in this paper assume that n>p holds in the data matrix. However, I think variable selection approaches are especially useful in the case of n<p. So, I want to know the theoretical results in such a case.

**Summary Of The Paper:**

This paper performs theoretical analyses of several variable selection approaches such as Lasso, Elastic net, SCAD, thresholded Lasso, forward selection, and forward backward selection. While the previous studies focused on the model selection consistency, this paper focused on the impassibleness of the model selection consistency based on Hamming errors. Its theoretical analyses show that SCAD has better performance than Lasso, Elastic net is worse than Lasso, and other approaches are superior to Lasso.

**Summary Of The Review:**

This paper is well-written.
Theoretical analyses for the case of n<p are needed in the paper.

---

### Official Review · Reviewer_darD · 2021-11-02

**Correctness:** 3
**Technical Novelty And Significance:** 2
**Empirical Novelty And Significance:** 1
**Recommendation:** 3
**Confidence:** 5

**Main Review:**

Strengths:
1. In general, the presentation of the paper is clear.
2. The theoretical results are carefully studied under the problem setup discussed in the paper.

Weakness:
1. The paper studied a very specific setting: "coefficients are iid drawn from a two-point mixture and that the Gram matrix is block-wise diagonal." The model is generated by three parameters. In my opinion, this setting is way simpler than the model settings studied in sparse linear regression literature. This may hurt the significance of the contribution.

2. The paper provides a long supplement file with proofs. However, to be honest, I did not spot many new technical contributions that are worth being commended as theoretical novelties. Besides, I am not sure if the results can be naturally generalized to more sparse linear regression settings.

3. The numerical studies are limited. Only a low-dimensional setting is considered. The figure at the bottom of page 9 lacks explanations. It is not clear to me what type of applications the results in the paper can be applied to.


**Summary Of The Paper:**

The paper compares several variable selection methods over block-wise diagonal designs. The performance is theoretically evaluated by the expected Hamming error. Some simulations are implemented to justify the empirical performance of the methods in the comparison.

**Summary Of The Review:**

Correctness:
Most of the theoretical results are well supported by the proofs in the supplement material. It is unclear if the pros and cons of the methods summarized in this paper can be generalized to more sparse linear regression setups.

Technical Novelty And Significance:
Sparse linear regression is a well-studied area. Although the authors have done a good job to deliver their theoretical results, I feel the technical novelty is limited. This paper focused on some specific settings. I also hold a conservative opinion towards its significance.

Empirical Novelty And Significance:
All the methods compared in the simulation study have mature algorithms to implement. The simulation is done under a low-dimensional setting (n=1000, p=300). The data generating processes are quite simple. I feel the novelty and significance of empirical studies are limited.

Flag For Ethics Review:
I do not foresee this research having any ethical issues.

---

### Official Review · Reviewer_YAB8 · 2021-11-03

**Correctness:** 4
**Technical Novelty And Significance:** 4
**Empirical Novelty And Significance:** 4
**Recommendation:** 8
**Confidence:** 4

**Main Review:**

Strong points

The article is a pleasure to read, with clearly exposed contributions. The adopted setting (“rare and weak signal” + “blockwise correlation”) is simple but very well motivated. This systematic study is comprehensive and several insights and intuitions are provided. The introduction of the Almost Full Recovery regime is interesting. Indeed, common model consistency properties can only separate the Full recovery regime from the other two.  Even though, strict model consistency does not hold, the Almost Full Recovery regime remains a useful regime to study for practitioners. The references to related works and similar settings are adequate and the work is well-placed in the literature. The experiments compliment the theoretical results.

Weak points

It is not stated explicitly in the theorems of Section 3 that the expected Hamming error is of shape $L_p p^{1-h(\vartheta, r, \rho)}$ contrary to the theorems of the supplements. This makes statements like “[…] with an ideal choice of $q$ that minimizes the exponent $E[H(\beta, \hat{\beta})]$” akward.

Typos:

(Main article)
- Figure and table without number or caption (page 9).
- Several typos in the references (inconsistent capitalisation of journal names, “et al.”).
- Legends in figures are too small to read on a printed article.

(Supplementary)
- Equation 7 : $\rho’$ (should be $\rho$).
- Lemma B.1, first bullet point : $\lambda’$ (should be $q$).
- Bottom of page 3 : “Figure 1 only dipicts”.
- Page 54 $(x_j, x_{j+2})$ (should be $(x_j, x_{j+1})$.

**Summary Of The Paper:**

The goal of the paper is to theoretically compare several variable selection schemes related to Lasso. To that end, the authors propose a comprehensive and visual methodology: phase diagrams. Under an idealised setting, they show that, depending on the strength and sparsity of the signal, variants of Lasso fall in one of the three following regimes, Full recovery, Almost Full Recovery, No Recovery. Their main results consist in the exact and non-asymptotic characterisation of those regimes. Once the phase diagrams are computed, the efficiency of algorithms can easily be compared. Most demonstrations rely on the fact that, under the “blockwise correlation” setting, methods can be  reduced to bivariate (Lasso-type) sub-problems.

**Summary Of The Review:**

This is a solid work that will be certainly be helpful to the community, and I recommend it for publication.

---

### Decision · Program_Chairs · 2022-01-20

**Decision:**

Accept (Poster)

**Comment:**

In the end, all reviewers agreed that this is a solid piece of work. However, there were also some doubts regarding the relevance of the block diagonal design and the underlying assumptions about the p/n ratio. The majority of the reviewers, on the other hand, had the impression that the positive aspects dominate the potential problems, and I also share this viewpoint. However, I'd like to encourage the authors to carefully address the points of criticism raised by the reviewers in their final version.